# Variations in the Compositions of Soil Bacterial and Fungal Communities Due to Microhabitat Effects Induced by Simulated Nitrogen Deposition of a Bamboo Forest in Wetland

**Weicheng Li [1,2,*]**, **Haiyan Sheng [3]**, **Desy Ekawati [4]**, **Yueping Jiang [5]** and **Huimin Yang [1]**

1   China National Bamboo Research Center/Key Laboratory of High Efficient Processing of Bamboo of Zhejiang Province, Hangzhou 310012, China; 18058785836@163.com
2   College of Life Sciences, Southwest Forestry University, Kunming 650224, China
3   Hangzhou Academy of Environmental Protection Science, Hangzhou 310005, China; laohaishy@163.com
4   Center for Social Economic Policy and Climate Change Research and Development/Forestry and Environment Research Development and Innovation Agency, Bogor 16610, Indonesia; lwc-1978@163.com
5   Xixi National Wetland Park Research Center for Ecological Science, Hangzhou 310030, China; leewis-1978@163.com
*   Correspondence: leewiscbrc@caf.ac.cn; Tel.: +86-130-6776-6917

**Abstract:** Although numerous studies have been published on nitrogen (N) deposition, little is known about its impact on microbial communities in wetland forests. Here, we used simulated nitrogen deposition (SND) to analyze the importance of differences in soil microhabitats in promoting the diversity of soil bacteria and fungi. We compared various levels of SND (control (CK), low N (N30), medium N (N60), and high N (N90)) and found that these were associated with changes in soil microhabitats. Additionally, SND affected soil pH, clay and sand content of the soil, and specific surface area (SSA). Bacteria and fungi responded differently to increased SND levels. The alpha diversity of bacteria decreased with an increased SND level, while fungal abundance, diversity, and community evenness reached their maximum values at the N60 threshold. Principal coordinates analysis (PCoA), nonparametric multivariate analysis of variance (PERMANOVA), and linear discriminant analysis (LDA) coupled with effect size measurements (LefSe) also confirmed that the bacterial composition was different at N90 compared to other levels of SND while that of fungi was different at N60. A higher discriminant level (LDA score ≥4) may be a valuable index of selecting indicator microbial clades sensitive to SND for wetland management. Further, an increased pH was associated with a greater abundance of bacteria and fungi. In addition, the volume contents of clay and SSA were negatively correlated with bacteria but fungi are associated with soil specific gravity (SSG). Overall, in a neutral soil pH environment, pH fluctuation is the main influencing factor in terms of bacterial and fungal abundance and diversity. The diversity of fungi is more dependent on the type and relative content of solid phase components in soil than that of bacteria, implying the presence of species-specific niches for bacteria and fungi. These results demonstrate that changes in SND can induce short-term microbial and microhabitat changes.

**Keywords:** *Phyllostachys violascens*; soil microhabitat; soil particle-size fractions; soil microbial diversity; soil pH value

## 1. Introduction

The mechanisms influencing nitrogen (N) deposition in wetland ecosystems are an important focus of study in research on global climate change [1,2]. Human activities have accelerated the

deposition of N in terrestrial ecosystems and, in many regions of the world, ecosystems and wetlands, in particular, display sensitivity to reactive N [3,4]. Several global N deposition prediction models have estimated that the subtropical regions of China will become the most severely impacted area in terms of global atmospheric N deposition in the coming decades [5–7]. Large amounts of N deposition alter the physicochemical properties of soil and affect plant–soil–microbial interactions [8,9]. Soil microbial communities play a vital role in the functioning of an ecosystem, as they are crucial for the establishment of a sustainable ecosystem that relies on healthy soil development and biological interactions [10,11]. In the past 30 years, although the responses of natural terrestrial ecosystems to increased N deposition have been extensively studied, the interactions within plant–soil–microbial communities under this condition are not well understood, especially in the urban wetland [6]. Although stringent measures that reduce N emissions to maintain ecosystem structure and function have been established in the past several decades, the Yangtze River Delta Metropolitan Area remains one of the more rapidly developing areas of industrialization and urbanization and one of the main areas of N deposition in China. Its dense plain river network and lakes form urban marginal wetlands, often with typical dike-pond systems [12]. Due to accelerated urbanization and the intensification of N deposition in the Yangtze River Delta, limited knowledge exists regarding changes in the soil microbial composition and structure in these wetlands.

Soil microorganisms can be affected by N deposition and features of the soil microhabitat, such as pH [13–15], bulk density and water-holding capacity [16,17], particulate size [18,19], and aggregate stability [20,21]. However, alterations in the composition of soil microbial communities due to changes in pH and particle size as a result of N deposition are not well understood despite the importance of microbial communities in the N cycle [15,22,23].

Soil pH varies both naturally and as a consequence of fertilization associated with silviculture and agronomic practices. It influences the soil microbial community structure and function due to the changes of microhabitat [24]. There is general agreement in the scientific community regarding ecosystem acidification caused by N deposition [15]; however, little is known about the mechanisms by which the soil microbial community copes with this acidification. A study of agricultural soils by Oehl et al. [25] revealed that different fungal species (saprophytic, ectomycorrhizal, and arbuscular mycorrhizal fungi) respond in different ways to variations in their microhabitat, with some species being more affected by soil pH and related parameters, such as soil texture, than others. The decrease in soil pH caused by elevated N inputs was also found to reduce the activity of soil microbes [13]. A watershed laboratory profile showed that the fungal internal transcribed spacer (ITS) was the most abundant at low pH and that bacterial 16S was the most abundant at high pH [24]; however, on average, simulated nitrogen deposition (SND) significantly decreased the ratio of fungi-to-bacteria [26]. N deposition also changes the composition and structure of the microbial community, and some fungi in microbial communities may be replaced by bacteria with higher N utilization [27]. N addition was shown to have no significant effect on microbial community structure along a soil pH gradient [27]; this may be due to the different effects of N forms and to nitrification on soil pH and, thus, fungal and bacterial composition and functions [24,26].

Particle-size fractionation, which allows the separation of soil organic matter with varying degrees of microbial alteration, might help elucidate microbe-mediated soil carbon (C) and N cycling characteristics [28,29]. Microbial acquisition of a substrate is compounded by relatively stable C and N levels [30], high nutrient and $O_2$ availability [31], and adsorption of extracellular enzymes to the clay and silt fractions but not to their aggregates [20]. As soil changes induced by N deposition are mainly the result of microbiological processes, different substrates and soil properties are selected by specifically adapted microbial communities [18,29]. A prior study demonstrated that changes in microbial communities depend on their microhabitats, with a greater abundance of bacterial and fungal genes observed in the soil fractions with larger-sized particles (>63 μm) compared to silt (2–20 μm) [28]. Poll et al. [32] revealed that coarse sand fractions are colonized mainly by a fungus-dominated community and a rather simple bacterial community whereas silt and clay fractions are colonized

by a complex bacterial community. Several studies have found that bacteria and fungi have a range of habitat requirements for soil with different particle sizes, which contain different amounts of C and N [18,25]. A physical separation of soils with different particle sizes is, therefore, necessary to retain their attached organic material and to ultimately provide a strategy for the characterization of conditions in different microhabitats.

Only a small number of studies have focused on the soil pH and microbial response to N deposition in urban wetlands [2,11]. In addition, the mechanisms of N deposition influencing soil microorganisms require more clarification. In this study, we sought to answer the following questions: (i) what are the main characteristics of soil microhabitats affected by N deposition, (ii) are the composition and structure of microbial communities affected by N deposition in the soil and, if so, which microbial species are most affected, and (iii) what are the main characteristics of microhabitat-microbial interactions that influence changes in microbial composition based on species-specific requirements following N deposition? We examined variations in the composition of the microbial community in topsoil (0–20 cm) and how they respond to soil pH and soil physical characteristics (soil water content, soil specific gravity, and bulk density) and to soil particle characteristics (clay, silt, sand, and specific surface area) under SND. Species-specific compositions of bacteria and fungi were evaluated by high-throughput sequencing of the 16S ribosomal RNA (rRNA) gene and ribosomal ITS, respectively. Overall, our study provides basic information for the forest management of N deposition and clarifies the features of species-specific microhabitats in urban wetlands.

## 2. Materials and Methods

### 2.1. Overview of the Study Area

The experimental site was Xixi National Wetland Park in the western suburb of Hangzhou, Zhejiang Province, China. Its total area is 16.15 km$^2$. The dike-pond is a typical ecosystem in the delta zone of the Yangtze River, where aquaculture in ponds and cultivation of economically important tree species on dikes has occurred for thousands of years, fertilizing the manure and pond sediment [33]. The region has a monsoon climate in the northern margin of the subtropical zone. The average annual temperature in the western area of Hangzhou is 16.4 °C, the frost-free period is 240 days, and annual precipitation is 1100–1600 mm. The daily average temperature, daily average humidity, and rainfall in the Xixi Wetland begins to rise from March to May, reaches a peak in June to August, and decreases in September to October [12]. The main soil type is paddy soil. A peat layer can be seen at an average depth of 60–100 cm.

### 2.2. Field Experiment and Soil Sampling

The experimental location was selected in Feijiatang Nature Reserve of the Xixi Wetland (120°03' E, 30°15' N). *Phyllostachys violascens* is the main economic species in the compound dike-pond production system of the Xixi Wetland due to its edible shoots. In recent decades, to better protect wetlands from being taken over by urban expansion, *Ph. violascens* forests have been left to grow naturally in the reserve. The dike width of the *Ph. violascens* forest ranges from 4 m to 8 m. A number of herbaceous plant species can be found in the undergrowth, including *Ophiopogon japonicus*, *Cardamine hirsuta*, *Pteris multifida*, *Torilis scabra*, *Mazus japonicus*, *Veronica peregrina*, *Lygodium japonicum*, *Oxalis corniculata*, and *Digitaria violascens*. The dike surface is 60–80 cm away from the water surface. The density, height, and diameter at breast height (DBH) of *Ph. violascens* are 2.6–3.1 stalks·m$^2$, 3.3–5.7 m, and 2.1–4.8 cm, respectively. The light density under the forest is 73.0–89.4% of full sunlight. Initial nutrient concentrations of experimental soils such as soil organic carbon, total nitrogen, and total phosphorus are 30.01–40.57 g·kg$^{-1}$, 1.28–4.07 g·kg$^{-1}$, and 57.66–112.08 mg·kg$^{-1}$.

Based on the actual amount of N deposition in subtropical China (30–37 kg·hm$^{-2}$ a$^{-1}$) [34] and its increasing trend [5,7], four levels of SND were derived: control (CK, 0 kg·hm$^{-1}$·a$^{-1}$), low N (N30, 30 kg·hm·hm$^{-2}$·a$^{-1}$), medium N (N60, 60 kg·hm$^{-2}$·a$^{-1}$), and high N (N90, 90 kg·hm$^{-2}$·a$^{-1}$).

Three 3 m × 3 m replicate plots were set up at each of the four SND levels for a total of 12 plots established in this study. There was an interval of 30–50 m between each plot as a buffer to prevent interaction.

SND spraying began in October 2016 and ended in September 2018. The annual N application rate was 12 equal applications, once per month. The specific application method was as follows: at the beginning of each month, quantitative ammonium nitrate ($NH_4NO_3$) was fully dissolved in 2.7 L pure water and uniformly sprayed on the surface of each plot of forest land using an electric back sprayer (equivalent to a 3 mm increase in annual precipitation). The control plot was sprayed with 2.7 L pure water without N [7]. In October 2018, after surface litter and impurities were removed, soil samples were collected from a depth of 0–20 cm by auger boring at three randomly selected points in each plot. Subsequently, a total of 36 soil samples were collected. Soil samples were stored at 4 °C in a sample box and transported to the laboratory within 6 h of collection. The samples were sieved (2 mm) for homogenization and removal of visible roots. Three equal portions of the sieved soil samples were retrieved. One portion of the sieved sample was stored at −78 °C for DNA extraction and subsequent 16S and ITS sequencing. Another portion was air-dried for particle-size analysis, and the remainder was used for the determination of soil pH and physical parameters.

### 2.3. Measurement of Soil Microhabitat Properties

Soil pH was measured in a 1:2.5 (*w/v*) soil:water extract using a glass LE410 pH electrode (FiveEasy™ pH meter; Mettler-Toledo AG, Zurich, Switzerland). The particle-size distribution was measured using a Malvern Mastersizer 3000 laser diffractometer (Malvern Instruments, Malvern, UK), which enabled rapid measurement of the volume percentage of particles in 100 size classes within a 0.01–3500 μm range, giving an output of specific surface area (SSA) data. Macroscopic traces of organic matter were removed from representative subsamples before being dampened by the dropwise addition of a standard chemical solution (40 g/L solution of sodium hexametaphosphate, $(NaPO_3)_6$, in distilled water) for aggregate dispersal. The mixture was fully reacted by agitation for 16 h and analyzed in a deionized water suspension with no sonication. A standard range of textural parameters was calculated, including the percentage of clay (<2 μm), fine silt (FS, 2–20 μm), coarse silt (CS, 20–63 μm), and sand class sizes (>63 μm) following the taxonomy guidelines of the U.S. Department of Agriculture [28]. The Malvern instrumentation was regularly calibrated using latex beads of known size.

The soil water content (SWC) was determined by the oven-dry weight method. The soil-specific gravity (SSG) was measured by the pycnometer method. Soil bulk density (BD) was determined by the core method with the undisturbed soil samples collected in 100 $cm^3$ steel cylinders and calculated as the oven-dry mass of the sample volume [12].

### 2.4. DNA Extraction, Polymerase Chain Reaction, and Illumina HiSeq Sequencing

DNA was extracted using a HiPure soil DNA kit B (Magen, Guangzhou, China). The hypervariable V3/V4 regions of the bacterial 16S rRNA gene were amplified using the forward primer 5′-CCTACGGGNGGCWGCAG-3′ and reverse primer 5′-GACTACHVGGGTATCTAATCC-3′ [35]. The ITS2 regions of the fungal rRNA gene were amplified using the forward primer 5′-GCATCGATGAAGAACGCAGC-3′ and reverse primer 5′-TCCTCCGCTTATTGATATGC-3′ [36].

The polymerase chain reaction (PCR) was set up in a volume of 25 μL as follows: 12.5 μL of 2 × KAPA HiFi HotStart ReadyMix (KAPA Biosystems, Wilmington, MA, USA), 1 μL of each primer, 5.5 μL of PCR-grade water, and 5 μL of DNA template. The PCR conditions were as follows: initial denaturation at 95 °C for 3 min, followed by 25 cycles (30 cycles for ITS) of denaturation at 95 °C for 30 s, annealing at 55 °C for 30 s, and extension at 72 °C for 30 s, with a final extension at 72 °C for 5 min. A specific tag sequence was added to the samples using the following PCR mixture (50 μL): 25 μL of 2 × KAPA HiFi HotStart ReadyMix, 2 μL of universal primer, 2 μL of barcode primer, 16 μL of PCR-grade water, and 5 μL of diluted template. The PCR conditions were as follows: initial denaturation at

95 °C for 3 min, followed by eight cycles of denaturation at 95 °C for 30 s, annealing at 55 °C for 30 s, and extension at 72 °C for 30 s, with a final extension at 72 °C for 5 min. The concentration of PCR products in each sample was measured using a NanoDrop ND1000 spectrophotometer (Wilmington, DE, USA) after gel extraction.

Library construction and Illumina HiSeq 2500 sequencing were performed by Shanghai Xiangyin Biological Technology Co., Ltd. (Shanghai, China).

*2.5. Data Processing and Statistical Analysis*

The obtained DNA sequence reads were trimmed using PANDAseq. After trimming, chimeras were removed using USEARCH 8.0 [37] and sample sequences were combined using QIIME_1.9.1 software [38]. An operational taxonomic unit (OTU) was defined as a set of DNA sequences with a similarity of >97%. OTUs were extracted using the Greengenes v13_08 database and classified by species. OTUs representing less than 0.005% of all sequences were removed before analysis. Diversity indexes were calculated using the QIIME software.

One-way ANOVA was used to test for significant differences among the different levels of SND. Multiple comparative analyses of OTU frequencies ($\alpha = 0.05$) were conducted in Data Processing System (DPS) (17.10 for Windows, Zhejiang University, Hangzhou, China) using the Benjamini–Hochberg method to control the familywise error [39]. Principal coordinates analysis (PCoA), redundancy analysis (RDA), nonparametric multivariate analysis of variance (PERMANOVA) of the microbial composition, as well as alpha-diversity type complexity analysis (including analysis of Chao1, Observed species (OS), and Shannon and Simpson index) were conducted at the phylum level using the "vegan" and "ggplot" package in R [40,41]. The fungi/bacteria ratio (F/B) was calculated based on absolute quantity (i.e., observed species) [18]. Linear discriminant analysis coupled with effect size measurements (LefSe) was conducted using the Kruskal–Wallis rank sum test in "stats" package in R [42].

## 3. Results

*3.1. Soil Microhabitats Affected by SND*

Among the increased SND levels, N60 and N90 were found to affect the soil pH of the *Ph. violascens* forest. Soil acidification was notable under these conditions (Table 1). Changes in SWC, SSG, and BD were not obvious. However, SND affected the volume percentage of clay and sand at the N90 level and N60 level, respectively. It had no effects on FS and CS individually, but the average value of the volume percentage of FS and CS reached a maximum at the N60 level and then decreased. Furthermore, the SSA reached its maximum at N60.

**Table 1.** Soil microhabitat characteristics and particle size fractions in the *Phyllostachys violascens* stands of different nitrogen addition levels: Values represent means ± SD (*n* = 9). Multiple comparison was adjusted with the Benjamini–Hochberg method.

|  | **CK** | **N30** | **N60** | **N90** |
|---|---|---|---|---|
| pH | 5.50 ± 0.16a | 5.43 ± 0.12a | 5.26 ± 0.12b | 5.14 ± 0.09b |
| Soil water content (wt.%) | 20.75 ± 1.55a | 19.89 ± 2.63a | 20.95 ± 2.11a | 21.95 ± 1.95a |
| Soil specific gravity | 2.47 ± 0.07a | 2.51 ± 0.09a | 2.54 ± 0.14a | 2.60 ± 0.27a |
| Bulk density (g·cm$^3$) | 1.31 ± 0.19a | 1.35 ± 0.20a | 1.33 ± 0.15a | 1.26 ± 0.14a |
| Clay (<2 μm, %) | 2.29 ± 0.28a | 2.28 ± 0.18a | 2.47 ± 0.13ab | 2.51 ± 0.11b |
| Fine silt (2–20 μm, %) | 29.68 ± 3.94a | 30.04 ± 4.21a | 33.22 ± 2.55a | 31.01 ± 4.11a |
| Coarse silt (20–63 μm, %) | 53.34 ± 4.16a | 55.47 ± 2.52a | 56.07 ± 2.64a | 54.93 ± 1.53a |
| Sand (<63 μm, %) | 14.69 ± 4.40a | 12.21 ± 5.61ab | 8.24 ± 2.45b | 11.55 ± 3.01ab |
| Specific surface area (m$^2$·kg) | 463.05 ± 30.42a | 468.39 ± 42.25ab | 512.63 ± 35.37b | 498.07 ± 58.09ab |

Values within a row with different letters are significantly different; *p*-value < 0.05.

### 3.2. Variations in the Structure and Composition of the Microbial Community under SND

Acidobacteria, Proteobacteria, and Planctomycetes were dominant at all SND levels, accounting for 58.30% to 70.25% of the total OTU abundance. The relative abundance of Acidobacteria increased with an increase in SND, and a significant effect was observed at the N60 and N90 levels (Table 2). Proteobacteria, Bacteroidetes, Chloroflexi, Actinobacteria, and Chlamydiae were not affected. Verrucomicrobia reached their maximum relative abundance at the N30 and N60 levels. Planctomycetes and Gemmatimonadetes decreased with an increase in SND. TM7 and OD1 showed sharp decreases at the N30, N60, and N90 levels, implying that TM7 and OD1 are sensitive to SND, which may disturb the established formation of intercellular metabolic networks. Nitrospirae reached their maximum at the N30-simulated nitrogen level, indicating that different species of bacteria differ in their responses to variations in the nitrogen concentrations and have their own unique species-specific demands. Ascomycota, Basidiomycota, and Zygomycota accounted for 44.03%–56.71% of the total OTU abundance of fungi. Ascomycota were not affected; Basidiomycota and Glomeromycota significantly increased with an increase in SND; Zygomycota decreased sharply at N60 and N90, while Chytridiomycota increased.

**Table 2.** Phylum distribution of dominant soil bacteria and fungi (relative abundance ≥1.0%) in the *Phyllostachys violascens* forests of different nitrogen addition levels: Values represent means ± SD ($n = 9$). Multiple comparison was adjusted with the Benjamini–Hochberg method.

| Kingdom | Phylum | CK | N30 | N60 | N90 |
|---|---|---|---|---|---|
| Bacteria | Acidobacteria | 30.48 ± 5.65c | 31.46 ± 2.49bc | 33.78 ± 5.00b | 38.05 ± 6.49a |
| | Proteobacteria | 16.78 ± 1.40a | 17.38 ± 2.18a | 18.29 ± 3.30a | 17.36 ± 3.28a |
| | Planctomycetes | 16.99 ± 3.19a | 14.28 ± 3.32b | 11.60 ± 2.07c | 11.20 ± 3.50c |
| | Bacteroidetes | 9.10 ± 2.24a | 8.81 ± 2.13a | 10.25 ± 3.75a | 8.71 ± 4.03a |
| | Verrucomicrobia | 5.02 ± 0.87c | 9.32 ± 1.98a | 9.03 ± 1.25a | 7.30 ± 1.71b |
| | Chloroflexi | 5.65 ± 1.02a | 5.55 ± 1.42a | 5.62 ± 2.41a | 5.65 ± 1.78a |
| | Actinobacteria | 1.88 ± 0.35a | 2.21 ± 0.71a | 2.20 ± 0.51a | 1.77 ± 0.46a |
| | Gemmatimonadetes | 1.47 ± 0.20a | 1.33 ± 0.40ab | 1.12 ± 0.41b | 1.02 ± 0.31b |
| | TM7 | 1.84 ± 0.30a | 0.73 ± 0.26b | 0.56 ± 0.15b | 0.62 ± 0.20b |
| | Nitrospirae | 1.63 ± 0.27b | 2.42 ± 0.91a | 1.78 ± 0.86b | 1.75 ± 0.75b |
| | Chlamydiae | 1.59 ± 0.72a | 1.24 ± 0.63a | 1.26 ± 0.34a | 1.30 ± 0.50a |
| | OD1 | 1.32 ± 0.57a | 0.38 ± 0.24b | 0.30 ± 0.14b | 0.22 ± 0.16b |
| Fungi | Ascomycota | 28.36 ± 10.25a | 32.15 ± 9.87a | 27.93 ± 6.97a | 30.73 ± 6.11a |
| | Basidiomycota | 8.13 ± 5.62b | 8.43 ± 7.58ab | 11.30 ± 7.58ab | 14.76 ± 7.58a |
| | Zygomycota | 10.70 ± 6.77a | 11.15 ± 7.75a | 3.14 ± 2.10b | 4.06 ± 2.25b |
| | Glomeromycota | 1.15 ± 1.32b | 2.81 ± 2.09a | 2.69 ± 1.53a | 3.34 ± 1.20a |
| | Chytridiomycota | 0.07 ± 0.06b | 0.05 ± 0.03b | 1.05 ± 0.47a | 0.61 ± 0.12a |
| | Unidentified | 49.90 ± 13.14a | 48.49 ± 13.71a | 54.46 ± 9.27a | 46.89 ± 7.19a |

Values within a row with different letters are significantly different; *p*-value < 0.05.

### 3.3. Variations in Microbial Alpha Diversity under SND

The alpha-diversity index of bacteria showed significant changes in N90 compared to CK and N30 (Table 3). At the N90 level, Chaol and OS had significant declines, indicating that bacterial communities were at a disadvantage in terms of abundance with an increase in SND. Chaol results also corroborated the OS index, indicating that CK was rich in diversity. The Shannon and Simpson index showed that diversity in CK was the highest while that in N90 was the lowest; this decreased with an increase in SND, indicating that the nitrogen level at N90 significantly reduced bacterial species diversity and community evenness. Behaviors of fungi differed from those of bacteria. Chaol revealed that there was no difference in the fungal richness at various levels of N. OB showed that the SND had no effect on the abundance of fungi. The Shannon and Simpson index revealed a significant difference between CK and N60, indicating that medium and high N levels increased diversity and community evenness. The observed F/B ratio increased with an increase in SND.

**Table 3.** Alpha diversity index of soil bacteria and fungi in the *Phyllostachys violascens* stands under different nitrogen addition levels: Values represent means ± SD (*n* = 9). Multiple comparison was adjusted with the Benjamini–Hochberg method.

| Kingdom | Index | CK | N30 | N60 | N90 |
|---------|-------|-----|-----|-----|-----|
| Bacteria | Chao1 | 3246.9 ± 583.4a | 3123.2 ± 249.4a | 2978.0 ± 271.4ab | 2597.4 ± 279.9b |
| | Observed species | 1817.8 ± 216.1a | 1785.9 ± 114.6ab | 1699.2 ± 127.3ab | 1642.9 ± 290.4b |
| | Shannon | 9.41 ± 0.26a | 9.37 ± 0.20a | 9.22 ± 0.25ab | 9.04 ± 0.39b |
| | Simpson | 0.9963 ± 0.0007a | 0.9959 ± 0.0008a | 0.9954 ± 0.0010a | 0.9943 ± 0.0015b |
| Fungi | Chao1 | 179.5 ± 20.7a | 180.4 ± 31.3a | 181.6 ± 26.7a | 170.4 ± 15.8a |
| | Observed species | 131.7 ± 19.2a | 138.2 ± 23.1a | 146.8 ± 19.1a | 141.5 ± 11.0a |
| | Shannon | 4.58 ± 0.69b | 4.90 ± 0.63ab | 5.28 ± 0.50a | 5.31 ± 0.26a |
| | Simpson | 0.8835 ± 0.0758b | 0.9173 ± 0.0399ab | 0.9375 ± 0.0304a | 0.9397 ± 0.0171a |
| | F/B | 0.0690 ± 0.0097b | 0.0685 ± 0.0122b | 0.0797 ± 0.0120ab | 0.0830 ± 0.0116a |

Values within a row with different letters are significantly different; *p*-value < 0.05.

### 3.4. Characteristics of Microbial Beta-Diversity under SND

A visualization method, PCoA, combined with PERMANOVA was applied to reveal similarities and differences in the data. The PC1 and PC2 dimensions explained 35.06% and 23.26%, respectively, of OTU information for the soil bacterial community (Figure 1a). Bacterial N30 displayed continuity with the soil sample information for CK and N60. There was no discernible gap between the data from N60 and N90. The results showed that the bacterial composition of the four SND levels was continuous and that the bacterial community structure was similar among plots of the same type. By PERMANOVA analysis, *p*-values of CK vs. N90 and of N30 vs. N90 for bacteria were 0.043 and 0.025, respectively, indicating that the bacterial composition of CK and N30 differed from that of N90. Such findings were consistent with the PCoA results. For the fungi, the PC1 and PC2 dimensions explained 41.51% and 12.87% of the OTU information, respectively (Figure 1b). The *p*-values of CK vs. N60, of CK vs. N90, and of N30 vs. N90 were 0.036, 0.014, and 0.033, respectively, revealing evident differences in the composition of the fungal communities and supporting findings of the PCoA analysis. These results implied that the selective pressure along N gradients is a strong driver of community phylogeny.

LEfSe detected differential clades at all nitrogen deposition simulation levels, which clarified statistically significant differences of diversity among the bacterial and fungal communities (Figure 2). At an LDA score of ≥4, LEfSe showed that the most notably differential clades were Planctomycetes (2) in the CK, Verrucomicrobia (3) at N30, and Actinobacteria (7) at N90 (Figure 2a). The cladogram also identified the Dothideomycetes class (3) and the Atheliales order (2) at N90, the Tremellomycetes class (3) and the unidentified class (3) at N60, and the Mortierellales order (2) at N30. There was no differential clade where the LDA score was ≥4 at N60 for bacteria and in the CK for fungi (Figure 2b). At an LDA score of ≥2, the number of bacterial discriminant clades decreased in soil from the CK to N90 levels (104, 48, 43, and 42) (Table 4); however, the number of fungi increased from the CK to N60 levels (72, 32, and 31) and increased dramatically at the N90 level (93). At the CK level, the LEfSe indicated Proteobacteria (22), Chloroflexi (10), Planctomycetes (10), Armatimonadetes (9), and Bacteroidetes (8) as the most differentially abundant bacterial taxa. It also revealed that, in contrast to CK, Proteobacteria (13, 12, and 13), Acidobacteria (8, 7, and 12), and Chloroflexi (5, 3, and 3) were commonly found at the N30–N90 levels. Verrucomicrobia (5) and Actinobacteria (4) in N30, Bacteroidetes (7) and Fusobacteria (7) in N60, and Spirochaetes (3) and Firmicutes (2) in N90 appeared as the main discriminant clades. Fungal LEfSe indicated that Ascomycota (41, 21, 18, and 69) and Basidiomycota (15, 7, 8, and 14) were the main discriminant clades, and they displayed sharp changes at the N90 level (Table 5). From these results, we inferred that bacterial groups differed along with fungal groups under SND, thereby suggesting that both bacteria and fungi are sensitive to N levels and that the fungal discriminant clades show an increased response to high N levels.

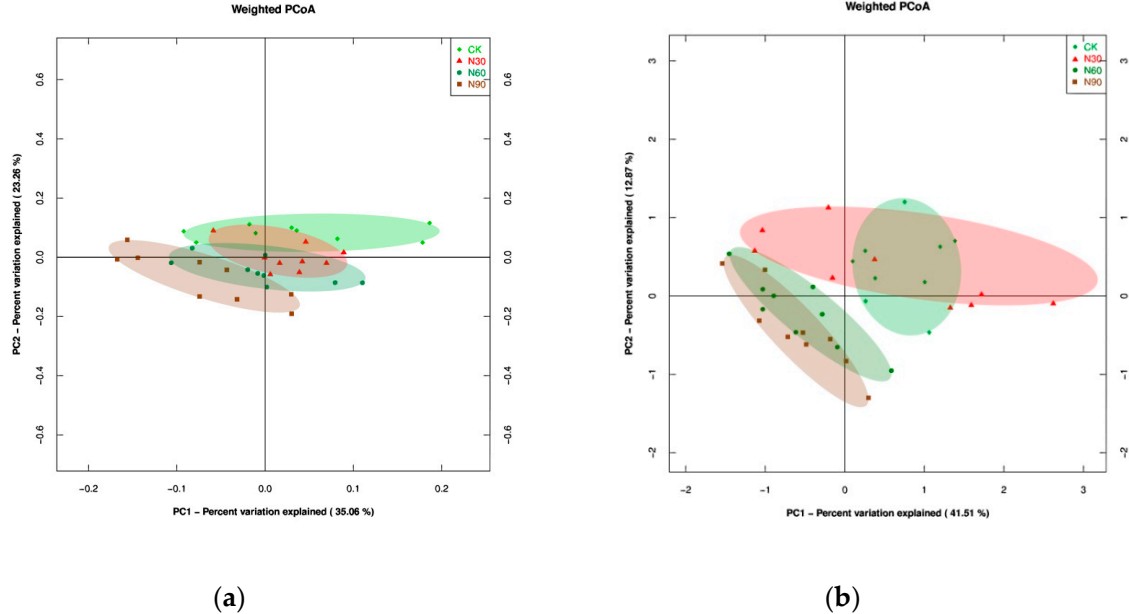

(a)                                                                 (b)

**Figure 1.** Principal coordinates analysis (PCoA) based on weighted calculated method and Bray–Curtis distances showing the changes of soil bacterial (**a**) and fungal (**b**) compositions (based on operational taxonomic unit (out) datasets of phyla) in the CK, N30, N60, and N90 levels. The transparent ellipse circle represents the distribution range of sampling plots. Light green diamond, green circle, brown square, and red triangle represent the soil samples collected from CK, N30, N60, and N90 plots, respectively, using the data profile of bacteria on phylum level (**a**).

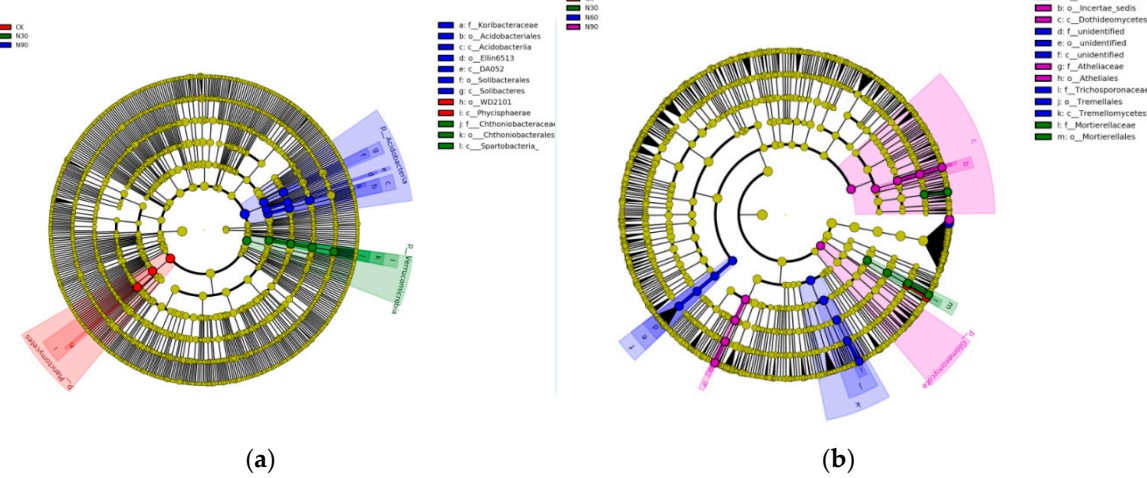

(a)                                                                 (b)

**Figure 2.** Discriminant taxa significantly retrieved by linear discriminant analysis (LDA) coupled with effect size measurements (LEfSe) analysis for bacterial (**a**) and fungal (**b**) communities at each different simulated nitrogen deposition levels: The cladogram shows the taxonomic representation of statistically consistent differences among CK, N30, N60, and N90 treatment soils in "bacteria" and "fungi" at a level of LDA score of ≥4 with the Kruskal–Wallis rank sum test. The circles radiated from inside to outside represent different taxonomic levels. Each small circle at a different taxonomic level represents taxa at that level. The diameter of the small circle is proportional to the relative abundance of OTU. Coloring Principle: The coloring of the nonsignificant species is yellow, and the different taxa are colored with groups, representing the microbial groups that play an important role in the group. The name of the taxa is represented in the legend on the right.

**Table 4.** Bacterial discriminant clades among CK, N30, N60, and N90 treatment soil are reported at a level of LDA score of ≥2.

| CK | | N30 | | N60 | | N90 | |
|---|---|---|---|---|---|---|---|
| Phylum/class | N. clade | Phylum/class | N. clade | Phylum/class | N. clade | Phylum/class | N. clade |
| Proteobacteria | 22 | Proteobacteria | 13 | Proteobacteria | 12 | Proteobacteria | 13 |
| Chloroflexi | 10 | Acidobacteria | 8 | Acidobacteria | 7 | Acidobacteria | 12 |
| Planctomycetes | 10 | Chloroflexi | 5 | Bacteroidetes | 7 | Chloroflexi | 3 |
| Armatimonadetes | 9 | Verrucomicrobia | 5 | Fusobacteria | 7 | Spirochaetes | 3 |
| Bacteroidetes | 8 | Actinobacteria | 4 | Chloroflexi | 3 | Firmicutes | 2 |
| Verrucomicrobia | 6 | Nitrospirae | 4 | Firmicutes | 3 | Bacteroidetes | 2 |
| Spirochaetes | 6 | Planctomycetes | 3 | Nitrospirae | 2 | Actinobacteria | 2 |
| Tenericutes | 4 | Armatimonadetes | 2 | Planctomycetes | 1 | Armatimonadetes | 2 |
| OD1 | 4 | Firmicutes | 2 | Gemmatimonadetes | 1 | Verrucomicrobia | 2 |
| Elusimicrobia | 3 | Bacteroidetes | 1 | | | WS3 | 1 |
| Acidobacteria | 2 | Gemmatimonadetes | 1 | | | | |
| Actinobacteria | 2 | | | | | | |
| WS6 | 2 | | | | | | |
| WS3 | 2 | | | | | | |
| BRC1 | 2 | | | | | | |
| Fibrobacteres | 2 | | | | | | |
| Gemmatimonadetes | 2 | | | | | | |
| GN02 | 2 | | | | | | |
| OP11 | 2 | | | | | | |
| TM7 | 2 | | | | | | |
| Firmicutes | 1 | | | | | | |
| Cyanobacteria | 1 | | | | | | |

**Table 5.** Fungal discriminant clades among CK, N30, N60, and N90 treatment soil are reported at a level of LDA score of ≥2.

| CK | | N30 | | N60 | | N90 | |
|---|---|---|---|---|---|---|---|
| Phylum/class | N. clade | Phylum/class | N. clade | Phylum/class | N. clade | Phylum/class | N. clade |
| Ascomycota | 41 | Ascomycota | 21 | Ascomycota | 18 | Ascomycota | 69 |
| Basidiomycota | 15 | Basidiomycota | 7 | Basidiomycota | 8 | Basidiomycota | 14 |
| Chytridiomycota | 4 | Zygomycota | 4 | Glomeromycota | 5 | Glomeromycota | 6 |
| Glomeromycota | 7 | | | | | Zygomycota | 4 |
| Zygomycota | 5 | | | | | | |

### 3.5. Relationship Between Soil Microhabitats and Microbial Compositions

Based on a Monte Carlo test ($F = 3.450$; $p = 0.002$, <0.01), RDA analysis showed that 98.3% of the information could be used to explain the relationship between bacterial composition and their environment. Soil bacteria (25 phyla) displayed a gradient correlation with soil pH, clay, and SSA. The positive correlation coefficient of soil pH on the first axis of environment was 0.767 (Figure 3a); the negative correlation coefficients of clay and SSA on the first axis of environment were −0.709 and −0.618, respectively, showing that bacterial communities were distributed along gradients of pH, clay, and SSA at the phylum level. In addition, Acidobacteria was positively correlated with clay, FS, and SSA and negatively correlated with pH by multiple comparisons (adjusted by the Benjamini–Hochberg method). Planctomycetes was positively correlated with pH, and Chlamydiae, Armatimonadetes, and Crenarchaeota were positively correlated with clay. AD3 was positively correlated with clay, FS, and SSA but negatively correlated with sand, while WS6 showed an inverse pattern compared to AD3.

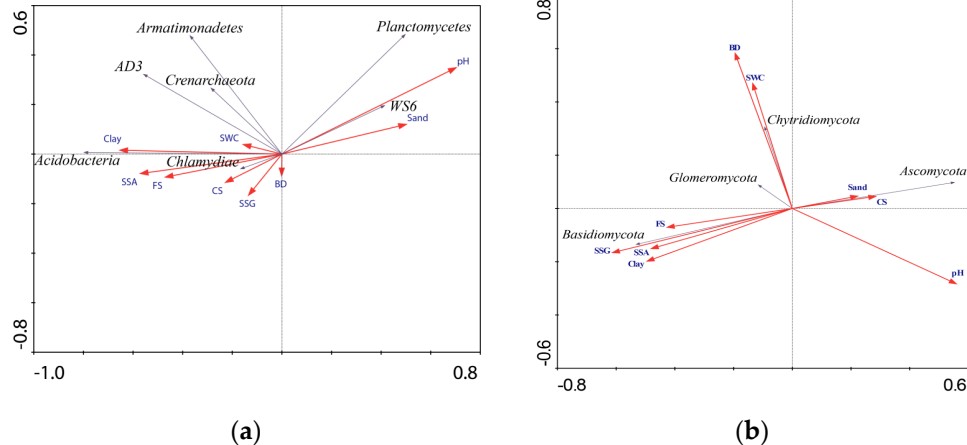

(a)　　　　　　　　　　　　　　　　　　　　　(b)

**Figure 3.** Biplot of redundancy analysis (RDA) of bacterial (**a**) (based on OTU datasets of 25 phylum) and fungal (**b**) (based on OTU datasets of 6 phylum) composition structure and soil microhabitat properties determined in soil samples collected from 36 sample plots in four N deposition simulation levels. The correlation coefficients on the first axis indicated that the bacteria were distributed along pH value, clay, and SSA gradients and that fungi were along pH value and SSG gradients in the *Phyllostachys violascens* forest. pH: soil pH value; SWC: soil water content; SSG: soil specific gravity; BD: bulk density; FS: fine silt; CS: coarse silt; SSA: specific surface area.

RDA analysis showed that 94.5% of the information could be used to explain the relationship between fungi and the environment using the Monte Carlo test ($F = 1.475$; $p = 0.060$, <0.1), showing that pH was positively correlated with the first axis (0.649) and that SSG was negatively correlated with the first axis (−0.612) (Figure 3b). Hence, there was a weak gradient correlation between fungi (6 phyla) and soil microhabitats, indicating that fungal communities are distributed along a gradient of pH and SSG. Meanwhile, Ascomycota was negatively correlated with clay, FS, and SSA and Basidiomycota was positively correlated with SSG. Glomeromycota was negatively correlated with pH, and Chytridiomycota was positively correlated with SWC.

## 4. Discussion

### 4.1. Main Characteristics of Soil Microhabitats

N deposition affects the soil pH as well as the chemical compositions of plant tissues [26], litter, and soil organic matter [43]. Consequently, it influences soil–plant–microbe feedbacks controlling substrate mineralization and the soil physical properties. N deposition can alter aspects of soil microhabitats, such as soil specific gravity, bulk density [44], and soil aggregate size [45], especially the particle size of microaggregates [28,30]. Also, Lu et al. reported that N deposition alteration on the soil microbial organisms is different from plants, with positive, negative, and no responses [46]. Our SND experiment revealed certain effects on microhabitat differences. The soil pH value ranged between 5.03 and 5.76. The pH value decreased with an increase in SND, which is consistent with the results of many prior studies [24,26,47]. Soil clay, sand, and SSA were also affected by SND. The volume percentage of clay increased with an increase in SND. The SSA reached its maximum at the N60 level. The results for sand, however, indicated that different SND levels affect the soil microhabitat differently through plant–microbial synergy. In addition, N slowly destroys soil aggregates and changes the activity of soil microorganisms [48].

### 4.2. Microbial Community Composition and Structure

The responses of microbial groups to N levels vary due to differences in their N uptake capacities [49]. For example, in plots with added N, N-demanding taxa may become more abundant than oligotrophic taxa, which eventually affects the metabolic capabilities of the soil microbial community

members [50]. Results of this study showed that bacterial phyla respond differently to an increase in SND levels, with their abundance either increasing, decreasing, or remaining unchanged, and that a similar response was seen for fungi. These complex responses can be expressed by the alpha-diversity index, which indicates differences among SND levels. The alpha index of bacteria differed at N90 as compared to other SND levels, while the alpha index of fungi first increased at N30 and N60 and then decreased at N90, suggesting that low–medium SNDs were beneficial for the development of the fungal community. There was a threshold effect of the N level on the fungal community structure (i.e., N60). The reason for this decrease may be that a high N level causes a hyperosmotic soil environment, thereby stressing the microbial community [51]. Therefore, the different changes of diversity indices implied that N level addition should consider the N uptake capacity, and the response of different microbial groups to N levels also need consider the impact of N inputs, including microbial physiology, soil mineralogy, and acidity. This is also reflected by F/B, which reveals the response of the soil microbial structure and function to soil conditions [18,52]. DeForest et al. [53] found that an increase in N deposition did not change the relative abundance of fungi and bacteria in a broad-leaved forest. Blaško et al. [54] also confirmed that, although bacteria and fungi decreased synchronously after SND, this did not affect F/B. On the contrary, Kou et al. [26] found that F/B decreased with an increase in SND in acidic soils (pH = 3.78) in a subtropical pine plantation. However, our data showed that F/B increased with SND levels, indicating that bacterial development was limited by the SND levels and that fungi increased the destruction of high-resistance carbon complexes [27]. This may be related to the time of SND and the background pH of the soil, as prior reports have argued that long-term N deposition had a greater negative impact on fungal biomass than that of bacteria, resulting in a decline in F/B [29,45,55]. Hence, after a short-term SND, soil microorganisms may still exist during the stress and adjustment phase, as indicated by the difference in the microbial trend of responses observed in this study compared to those observed for the long-term N deposition.

The interrelationships of environmental parameters among taxonomic subgroups may play a regulating role [20]. Beta diversity results of PCoA and PERMANOVA also showed that N30 had no significant effect on the community composition of bacteria and fungi; however, bacteria displayed a significant difference at N90, while the difference became evident for fungi at N60. As a whole, bacteria had a lower sensitivity to N than fungi. This may be related to the resource utilization strategy (r vs. K-strategist) of individual populations within subgroups (higher taxa) [56]. Meanwhile, the significant decrease in the number of bacterial discriminant clades by LEfSe indicated that microbial composition was sensitive to N application. Both bacterial and fungal changes in phylum/class discriminant clades responded to changes in alpha diversity, especially those in the fungal community at the N60–N90 levels, thereby implying that the abundance of soil microorganisms below the class level varied with SND levels. This is consistent with the findings of Mapelli [57], which showed that bacterial community development in bulk soil vs. rhizosphere soil coincides with soil fertility gradients. In addition, we found that several clades, including the bacterial clades Planctomycetes, Verrucomicrobia, and Actinobacteria and the fungal clades Dothideomycetes, Atheliales, Tremellomycetes, and Mortierellales had a higher discriminant level (LDA scores of ≥4). This may be helpful for the future use of microbial indicators for improved management of wetland N deposition.

### 4.3. Main Characteristics of Microhabitat–Microbial Interactions under SND

Changes induced by SND in microhabitats involve a series of tortuous physical networks, which determine the flow of substrates and solutes over space and time, influencing microbial interactions [20]. The decrease in soil pH caused by N deposition is one of the main factors affecting the change in soil microbial composition and functional diversity [6]. Liu et al. [13] found that the response of soil microorganisms to SND was negative in a temperate steppe, aligning with our results that the abundance and diversity of bacterial species decrease with an increase in pH. This was also consistent with the observation that the soil bacterial community tends to be more homogeneous under neutral pH conditions and has a higher richness and diversity [58]. Bååth et al. [59] showed

that there is a significant positive correlation between pH and microbial composition in the pH range of 3–9, which is consistent with the results of our study demonstrating that bacteria and fungi were positively distributed along the pH gradient. Similar results were obtained in a forested riparian zone by Fisher et al. [24], where fungal ITS was greatest at a low pH and bacterial 16S was the highest at a high pH. This is because N deposition may also indirectly change the availability of soil exchange base and reduce the buffering capacity of soil to inhibit the activity of microorganisms by reducing the soil pH [13]. However, the abundance, diversity, and community evenness of fungi reached a maximum at N60, which could be explained by (1) a large number of unidentified groups (OTU > 50%); (2) fungal development reaching the N threshold beyond the tolerance of fungi, which inevitably causes a decline; and/or (3) changes in other physical characteristics of microhabitats caused by nitrogen-dominated soil–microbial interactions.

Understanding the relationship between microhabitats and the soil microbiome is an important consideration to achieve sustainability in an ecosystem [20]. Kabir et al. [60] found that *Azospirillum brasilense* was sensitive to the clay content in cultivating perennial grass, while Fox et al. [20] showed that aggregate size in the form of macroaggregate (>250 μm), microaggregate (<250 μm), and silt and clay (SC, <53 μm) fractions had a significant effect on the bacterial community at both the phylum and family taxonomic levels. In addition to this, we found that not only the volume of clay but also SSA were negatively correlated with bacterial distribution. Fungi have minor effects on the size of particles because a lower clay content is key for fungal colonization [19]. In fact, fungal distribution was negatively correlated with SSG, which mainly depends on the type and relative content of solid phase components in soil. This indirectly corroborates the finding that the structural diversity of fungi responds more strongly to various particle size fractions than does bacteria [18].

Indeed, previous studies suggest that communities of arbuscular mycorrhizal fungi (AMF) are structured by soil properties [16,19,61]. Ordination of soil characteristics (mainly pH, silt, and sand) improved the predictions of AMF abundance but was not a significant contributor [62]. Soil type and texture correlated with AMF communities [63], possibly because AMF can increase plant uptake of nutrients [60]. In our study, Glomeromycota (AMF phyla) was negatively correlated with pH but not soil texture. This result indicated that the influence of soil pH is greater than that of other soil texture parameters on the microbial community. Meanwhile, different phyla of bacteria and fungi were found to correlate with clay, FS, sand, SSA, SSG, and SWC to varying degrees, suggesting that different microbial communities require different types of microhabitats for development and colonization, including an appropriate specialty niche based on particle type and size [18,45].

## 5. Conclusions

In summary, our study showed that bacteria and fungi respond differently to SND levels, which are associated with a series of microhabitat changes (i.e., in soil pH, clay content, sand content, and SSA). Bacterial abundance in taxonomic groups below the class level declined with an increase in SND levels, with a significant difference found at the highest SND level. In contrast, the threshold of fungal responses was found to occur at a medium SND level. Species-specificity was found to be influenced by the pH value for both bacteria and fungi. Bacteria demonstrated a negative correlation with clay content and SSA, while fungi were negatively correlated with SSG. These findings suggest that the compositions of the bacterial and fungal communities had a significant interaction with different types of microaggregates. Thus, the selective effects of microhabitats induced by SND strongly contribute to niche separation. Based on these results, we recommend that the observation of microbial niche changes and indicator clades should be used as an important component in wetland forest management programs.

**Author Contributions:** Methodology, W.L. and Y.J.; validation, W.L. and H.S.; formal analysis, H.S.; investigation, W.L. and H.Y.; resources, Y.J.; writing—original draft preparation, W.L.; writing—review and editing, W.L. and D.E.

**Funding:** This research was funded by Central Nonprofit Research Institution of Chinese Academy of Forest (CAF), grant number CAFYBB2017MA024

**Conflicts of Interest:** All the authors declare that they have no conflict of interest.

## Abbreviations

| | |
|---|---|
| Simulated nitrogen deposition | SND |
| Control | CK |
| low N | N30 |
| medium N | N60 |
| high N | N90 |
| internal transcribed spacer | ITS |
| ribosomal RNA | rRNA |
| soil water content | SWC |
| soil specific gravity | SSG |
| specific surface area | SSA |
| observed species | OS |
| fungi/bacteria | F/B |

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
