# Peer review of "Variations in the Compositions of Soil Bacterial and Fungal Communities Due to Microhabitat Effects Induced by Simulated Nitrogen Deposition of a Bamboo Forest in Wetland"

_forests, doi:10.3390/f10121098_

Round 1

Reviewer 1 Report

General comment

This manuscript deals with the study, in the field, of the effects of N deposition on bacterial and fungal communities of wetland soil. In my opinion, the paper is well written, can be readily understood and the methods are sound. The results are interesting and the topic is important. Thus, I recommend the paper to be accepted pending the following minor revisions.

Minor revisions

Line 23

“increase…” of?; or perhaps “increased” instead of increase

Line 28

LDA score; authors should write what LDA stands for in full extent also

Line 65

“composition of soil microorganisms” perhaps change to “composition of soil microbial communities”

Line 105

“Only a small number of studies have focused on the soil pH and microbial response to N deposition in urban wetlands.”

Please give examples of these studies.

Line 144

Parentheses are missing in DBH.

Lines 187 and 189

Authors should add references for the use of these primers.

Line 218

Which R package? Please specify and cite accordingly.

Line 220

Again, please specify and cite the R package used. 

Line 231

On the figure legend, the species name should be written in full. Perhaps the p-value could also be mentioned again (although it is referred in the methods).  Maybe it should be mentioned that different letters represent statistical differences.

Line 251

Table caption - please see previous comment regarding Table 1 caption (comment regarding line 231)

Line 280

Typo in PERMNOVA; -> PERMANOVA

Line 288

“by the conditions of the heterogeneity soil”, please rephrase, it is not clear.

Line 291

“Principal”

Line 296

It was not necessary to reverse the symbols to make a distinction, since these are two separate figures. It might be confusing.

Line 297

“PCoA indicated that…” Describing results in figure captions is useful in many cases; however, since this was not done for the other figures, I do not see the point; the sentence might be included in the text.  

Fig.2

The legend embedded in the figure is hard to read (too small). Legend for “bacteria” has a typo in “Acidobacteriia” -> Acidobacteria.

Line 321/Line 324

Bacterial “left” and fungal “right” is correct; right “bacteria” and left “fungi” is not correct; please check if the position of figures in the final version matches the captions.

Line 332

Correct to “The name of the taxa is represented” (“is” is missing)

Line 409

“evidence” -> “evident”?

Lines 29, 340, 351, 418, 462

The word “Further” was used quite a few times to start sentences; using a few synonyms might improve the text. 

Author Response

Reviewer: 1

Minor revisions

Line 23

“increase…” of?; or perhaps “increased” instead of increase

Answer: The word was replaced with “increased” on line 23.

Line 28

LDA score; authors should write what LDA stands for in full extent

Answer: The full extent standing for LDA was offered on line 26. “linear discriminant analysis (LDA)”.

Line 65

“composition of soil microorganisms” perhaps change to “composition of soil microbial communities”

Answer: Yes, we agreed with it. The phrase was changed on line 65.

Line 105

“Only a small number of studies have focused on the soil pH and microbial response to N deposition in urban wetlands.” Please give examples of these studies.

Answer: The reference was offered on line 106.

Line 144

Parentheses are missing in DBH.

Answer: Parentheses were added on line 144.

Lines 187 and 189

Authors should add references for the use of these primers.

Answer: Reference of primers was added on line

Line 218

Which R package? Please specify and cite accordingly.

Answer: The “vegan”, “ggplot2” packages and the related references were added on lines 225-226.

Line 220

Again, please specify and cite the R package used. 

Answer: The “stats” packages was added on line 229.

Line 231

On the figure legend, the species name should be written in full. Perhaps the p-value could also be mentioned again (although it is referred in the methods).  Maybe it should be mentioned that different letters represent statistical differences.

Answer: Thanks for this suggestion. The full species name was offered. The p-value was mentioned again, and the note of different letters representation was added throughout of the paper.

Line 251

Table caption - please see previous comment regarding Table 1 caption (comment regarding line 231)

Answer: The comments were accepted and all of the table captions were revised accordingly.

Line 280

Typo in PERMNOVA; -> PERMANOVA

Answer: The wrong abbreviation was revised.

Line 288

“by the conditions of the heterogeneity soil”, please rephrase, it is not clear.

Answer: The confusing phrase “by the conditions of the heterogeneity soil” was deleted.

Line 291

“Principal”

Answer: The missed word “Principal” was added.

Line 296

It was not necessary to reverse the symbols to make a distinction, since these are two separate figures. It might be confusing.

Answer: The confusing caption sentence was deleted, and symbols in the right figure were made as the same as the left figure.

Line 297

“PCoA indicated that…” Describing results in figure captions is useful in many cases; however, since this was not done for the other figures, I do not see the point; the sentence might be included in the text.  

Answer: The sentence was deleted in the caption.

Fig.2

The legend embedded in the figure is hard to read (too small). Legend for “bacteria” has a typo in “Acidobacteriia” -> Acidobacteria.

Answer: Thanks for this suggestion. We can provide the bigger figure if the editor wanted. “Acidobacteriia” is the class level under phylum Acidobacteria. So the legend was showed “c_ Acidobacteriia”. Here “c_” means class level.

Line 321/Line 324

Bacterial “left” and fungal “right” is correct; right “bacteria” and left “fungi” is not correct; please check if the position of figures in the final version matches the captions.

Answer: We corrected the wrong information.

Line 332

Correct to “The name of the taxa is represented” (“is” is missing)

Answer: The sentence was corrected.

Line 409

“evidence” -> “evident”?

Answer: We revised this word.

Lines 29, 340, 351, 418, 462

The word “Further” was used quite a few times to start sentences; using a few synonyms might improve the text. 

Answer: We have changed the expression mode.

Reviewer 2 Report

General comments          

Understanding of what controls the microbial communities in soils is important and this study addresses what happens under different nitrogen deposition scenarios. This is a timely and important issue. Unfortunately, the paper is too narrowly written and not easily accessible except to a narrow field of experts in the area. Results should be explained more broadly. For example, what do the different indices of diversity tell us. That different levels of nitrogen additions change the composition of the microbial community is not surprising but has this consequences for functions, e.g. carbon and nitrogen cycling in the soil.

Specific comments

How can nitrogen additions change soil texture? Such changes seem to be important for the interpretation of the results.

Is it meaningful to present SSG values without defining the reference substrate?

Line 255. The N90 group, group is confusing here and at other places. Does it mean anything other than treatment?

Author Response

Reviewer: 2

Results should be explained more broadly. For example, what do the different indices of diversity tell us?

Answer: We revised the explanation on line 409-413. “So the different changes of diversity indices implied that N levels addition should consider the N uptake capacity, and the response of different microbial groups to N levels also need consider the impact of N inputs, even including microbial physiology, soil mineralogy and acidity.”

That different levels of nitrogen additions change the composition of the microbial community is not surprising but has this consequences for functions, e.g. carbon and nitrogen cycling in the soil.

Answer: Thanks for this suggestion. With reference to KEGG, we have also sequenced metagenome and sorted out the functional genes referring to carbon and nitrogen cycle. Anyway, the metagenome dataset is not suitable for this microhabitats paper. We will discuss this topic with the soil biochemical properties in another manuscript later.  

Specific comments

How can nitrogen additions change soil texture? Such changes seem to be important for the interpretation of the results.

Answer: Thanks for this suggestion. We gave the interpretation of change soil texture by nitrogen additions on line 381-387. And we added the interpretation of the results on line 382-383. “Consequently, it influences soil-plant-microbe feedbacks controlling substrate mineralization, and then the soil physical properties.”

Is it meaningful to present SSG values without defining the reference substrate?

Answer: The reference substrates were offered on line 146-147. “Initial nutrient concentrations of experimental soils such as soil organic carbon, total nitrogen and total phosphorus are 30.01-40.57 g·kg-1, 1.28-4.07 g·kg-1, 57.66-112.08 mg·kg-1.”

Line 255. The N90 group, group is confusing here and at other places. Does it mean anything other than treatment?

Answer: The confusing phrase was revised.

Round 2

Reviewer 2 Report

No comments on the revised manuscript